# Rendering NK Cells Antigen-Specific for the Therapy of Solid Tumours

**DOI:** 10.3390/ijms26136290

**Published:** 2025-06-29

**Authors:** Carina A. Doeppner, Amanda Katharina Binder, Franziska Bremm, Niklas Feuchter, Jan Dörrie, Niels Schaft

**Affiliations:** 1Department of Dermatology, Universitätsklinikum Erlangen, Friedrich-Alexander-Universität Erlangen-Nürnberg, 91054 Erlangen, Germany; carina.doeppner@extern.uk-erlangen.de (C.A.D.); amanda.binder@uk-erlangen.de (A.K.B.); franziska.bremm@uk-erlangen.de (F.B.); niklas.feuchter@uk-erlangen.de (N.F.); jan.doerrie@uk-erlangen.de (J.D.); 2Comprehensive Cancer Center Erlangen European Metropolitan Area of Nuremberg (CCC ER-EMN), 91054 Erlangen, Germany; 3Deutsches Zentrum Immuntherapie (DZI), 91054 Erlangen, Germany; 4Bavarian Cancer Research Center (BZKF), 91054 Erlangen, Germany; 5Comprehensive Cancer Center Würzburg Erlangen Regensburg Augsburg (CCC-WERA), 91054 Erlangen, Germany

**Keywords:** chimeric antigen receptors (CAR), CAR-NK cells, immunotherapy, solid tumours, tumour microenvironment, combination therapy, clinical application, adoptive cell therapy

## Abstract

Cancer remains one of the leading causes of death worldwide. New treatments like immunotherapy—especially checkpoint inhibitors and CAR-T cell therapy—have improved outcomes for some patients. However, these therapies often struggle to treat solid tumours effectively. Natural killer (NK) cells are part of the immune system and can naturally detect and destroy cancer cells without previous adaption. Scientists are now enhancing these cells by adding special receptors, called CARs (chimeric antigen receptors), to help them better recognize and attack cancer, an approach originally developed for T cells. CAR-NK cell therapy has some advantages over CAR-T therapy. It tends to cause fewer severe side effects, such as strong immune reactions or off-target effects in healthy tissues. Within some limitations, the allogenic use of CAR-NK cells is possible, as these cells exert less graft-versus-host activity. Such CAR-NK cell products can be produced in larger quantities and stored, making treatment more accessible. Still, there are challenges. It can be difficult to create enough modified NK cells, and the tumour microenvironment can block their activity. This review highlights recent progress in CAR-NK therapy, including early lab and clinical research. It also explores ways to improve these treatments and how they might work alongside other cancer therapies to help more patients in the future.

## 1. From NK-Cell Classification to CAR-NK Therapy: Addressing the Challenges of Immunotherapy in Solid Tumours

Cancer is one of the leading causes of mortality worldwide. Even though immunotherapies such as checkpoint inhibitors and CAR-T cells have revolutionized treatment, challenges remain, especially in solid tumours. Natural killer (NK) cells, an essential part of the innate immune system, have emerged as a promising candidate for cancer immunotherapy due to their ability to identify and destroy cancerous cells without prior antigen sensitization. Unlike T cells, NK cells rely on a balance of activating and inhibitory signals. Their ability to detect and eliminate tumour cells through mechanisms such as perforin- and granzyme-mediated cytotoxicity and antibody-dependent cellular cytotoxicity (ADCC) makes them attractive for cancer immunotherapy [1,2,3].

To increase the specific recognition of tumour cells and thus improve the efficacy of NK-cell therapy, particularly against solid tumours, researchers are currently focusing on equipping NK cells with tumour antigen-specific receptors. Chimeric antigen receptor (CAR)-modified NK cells are designed to recognize tumour-associated antigens with high specificity. Unlike traditional NK cells, CAR-NK cells can be tailored to selectively target tumour cells that might otherwise evade immune detection.

Despite their potential, several challenges remain, such as low transduction efficiency, suboptimal expansion techniques, and the immunosuppressive tumour microenvironment (TME), which can hinder CAR-NK-cell function. Nonetheless, CAR-NK cells represent a promising alternative to existing cancer therapies, with ongoing research to overcome these limitations [4].

This review provides a comprehensive analysis of recent advancements in antigen-specific NK-cell therapy for solid tumours. It summarizes key breakthroughs in CAR-NK-cell engineering, as well as their preclinical and clinical applications. Further, it discusses therapeutical implications and challenges, while addressing ongoing debates, such as the combination of CAR-NK cells with existing tumour therapies and strategies to navigate the complex TME. Given the rapid advancements in antigen-specific NK-cell engineering, a comprehensive review of these advancements is crucial to understanding their clinical potential and guiding future research directions.

### 1.1. NK Cell Classification and the Role of Non-Manipulated NK Cells in Cancer

NK cells are derived from multipotent haematopoietic stem cells and develop through common lymphoid progenitors and NK/T-cell precursors. Unlike T cells, which mature in the thymus, NK cells mature in the bone marrow [5]. While both NK and T cells share a common progenitor, they differ significantly in their roles, mechanisms of action, and their activation requirements. Although both are lymphocytes, one obvious distinction is that NK cells belong to the innate immune system and T cells to the adaptive immune system. Further, the main difference is that NK cells do not require antigen presentation or prior sensitization to recognize and destroy their targets, unlike T lymphocytes. Instead, NK cells rely on a complex balance of inhibitory and activating receptors to identify and eliminate abnormal cells, particularly cancer cells [1,3,6]. These receptors are germline-encoded and thus do not require “V(D)J” recombination for their diversity and functions like T cells, further distinguishing NK cells from T cells [7].

Phenotypically, NK cells are characterized by the expression of neural cell adhesion molecules (NCAM/CD56) and the absence of CD3 and T-cell receptors. Additionally, the absence of c-kit (a marker present in innate lymphoid cells type 3, ILC3) further differentiates NK cells [8]. NK cells are mainly located in the lymph nodes and the peripheral circulation, comprising approximately 2% of the leukocytes in the blood [9]. Based on the characterization of NK cells in the peripheral blood, human NK cells are conventionally divided into two major subpopulations (CD56^bright^CD16^dim^ and CD56^dim^CD16^bright^), where the CD56^bright^CD16^dim^ ones were believed to be less mature and more potent cytokine producers and the CD56^dim^CD16^bright^ ones to be more mature and cytotoxic ones (Figure 1) [5]. However, recent RNAseq-based data suggest that human NK cells exhibit significant heterogeneity and can be classified into three main subsets: NK1 (high protein expression of CD16, CX3CR1, CD161, β7-integrin, and CD38), NK2 (high expression of CD56, CD27, CD44, CD54, CD45RB, NKG2D, and NKp46 and little or no expression of CD16 and CD57 at the protein level), and NK3 (protein expression profile includes CD16, CD57, CD271 (NGFR), CD2, CD18, CD49d, and inhibitory killer cell immunoglobulin-like receptors (KIRs) (CD158e, CD158b), with lower expression levels of CD56, NKp30, NKp46, CD161, and CD122) (Figure 1). Each subset exhibits unique transcription factors, metabolic traits, and cytokine responses, influencing their distinct roles in responses against tumours and infections [10].

**Figure 1 ijms-26-06290-f001:**
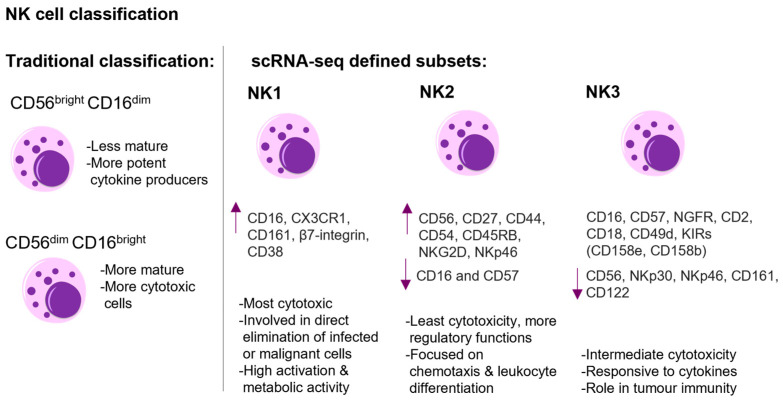
The figure shows the classification and schematic representation of human NK cell heterogeneity. On the left side, the traditional classification into CD56^bright^CD16^dim^ and CD56^dim^CD16^bright^ subsets with their respective functional attributes are shown. On the right side, the NK cell classification defined by scRNA-sequencing is shown. The three main subsets—NK1, NK2, NK3—are presented. The NK1 subset exhibits potent cytotoxicity, marked by high expression (arrow up) of CD16, CX3CR1, CD161, β7-integrin, and CD38. NK2 cells display high expression of CD56, CD27, CD44, CD54, CD45RB, NKG2D, NKp46, and little or no expression (arrow down) of CD16 and CD57 at the protein level. NK2 cells exhibit lower cytotoxic potential compared to NK1 and NK3 and are more focused on regulatory functions. NK3 cells demonstrate intermediate cytotoxicity, and the protein expression profile includes CD16, CD57, CD271 (NGFR), CD2, CD18, CD49d, and inhibitory killer cell immunoglobulin-like receptors (KIRs) (CD158e, CD158b), with lower expression levels of CD56, NKp30, NKp46, CD161, and CD122 [5,10]. This figure was generated with Motifolio.

NK-cell activation and function depends on a finely tuned interaction between activating and inhibitory receptors and their respective ligands. This balance determines whether NK cells engage in cytotoxic activities against a target cell (summarized in Figure 2) [11].

#### 1.1.1. Activating NK Receptors

Activation signals are initiated and transmitted when the activating receptors on NK cells interact with their specific ligands on target cells. Coordinated receptor interactions ensure a synergistic and potent cytolytic response against abnormal cells, such as tumour and/or virally infected cells [12] (Figure 2). Important activating receptors are the natural cytotoxicity receptors (NCRs) [13], which include NKp44 [14], NKp46 [15], and NKp30. NKp30 directly targets the tumour cell ligand B7-H6 of the B7 family, inducing NK-cell activation and toxicity [16,17]. It acts in a similar way to CD28H (the CD28 homologue), which binds to B7-H7 in T-cell signalling [18]. A deficiency in NKp46 has been linked to impaired clearance of T-cell lymphoma, melanoma, and lung metastasis [6]. Another activating receptor is the DNAX accessory molecule (DNAM-1), also known as CD226, which is a type I transmembrane glycoprotein. The interaction of DNAM-1 with ligands such as CD155 and CD112 has been shown to mediate the production of interferon gamma (IFNγ) and trigger NK-cell-mediated cytotoxicity. The expression of these ligands is predominantly elevated in solid tumours, with minimal expression observed in normal tissues [19]. The natural killer group 2 member D (NKG2D), a type II transmembrane and C-type lectin-like receptor, is associated with the DNAX-activating protein 10 (DAP10) to transmit activating signals. The ligands of NKG2D are the HLA (human leukocyte antigen) homologues ULBP (UL16-binding proteins) and HLA class I-related molecules (MIC)A/B, which are induced by genotoxic stress or DNA damage [20], oxidative stress [21], and viral infection [22]. Since these HLA homologues are generally expressed on epithelial-derived tumour cells, including ovarian cancer and colon cancer, the NKG2D pathway is able to regulate tumour initiation and progression. However, it should be noted that tumour cells can produce soluble NKG2DL, which can lead to a downregulation of NKG2D. This has been associated with breast cancer lymph node metastasis, melanoma, neuroblastoma, prostate cancer, and kidney cancer [19]. Further activating receptors on natural killer cells are 2B4, DNA1, LFA-1, CD28H, CD2, and NKG2C [23].

**Figure 2 ijms-26-06290-f002:**
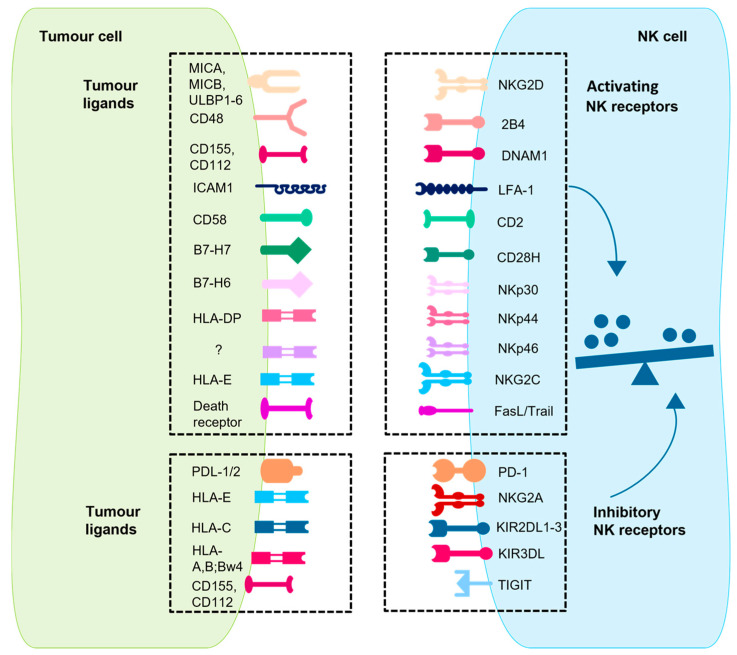
The interaction network between tumour cells and NK cells is shown. This is a schematic representation of key activating and inhibitory receptors on mature NK cells and their corresponding ligands on tumour cells. The interaction between these receptors and ligands—either constitutively expressed or upregulated during tumour transformation—regulates NK cell activation. The balance of these signals determines whether NK cells initiate cytotoxic responses or remain inhibited. CD28H: CD28 homologue; DNAM1: DNAX accessory molecule 1; HLA: human leukocyte antigen; ICAM1: intercellular adhesion molecule 1; KIR2DL: killer cell immunoglobulin-like receptor 2DL; LFA-1: leukocyte function-associated molecule 1; MIC: major histocompatibility complex class I polypeptide-related sequence; NKG2: natural killer group 2; PD-1: programmed cell death protein 1; PDL-1/2: PD-1/2 ligand; TIGIT: T cell immunoreceptor with immunoglobulin and ITIM domains; ULBP: UL16 binding protein [23,24]. This figure was generated with Motifolio.

#### 1.1.2. Inhibitory NK Receptors

There are two broad categories of inhibitory receptors on NK cells: HLA-specific receptors and non-HLA-specific receptors [25] (Figure 2). The HLA-specific inhibitory receptors, including KIRs [26] and NKG2A, primarily regulate NK-cell function through interactions with HLA class I molecules. These are critical for the recognition of healthy cells, thereby preventing activation against self-cells. Of particular significance are the inhibitory KIRs (iKIRs), KIR2DL1, KIR2DL2, and KIR2DL3, which are immune 1 globulin-like, extracellular surface receptors with an immunoreceptor tyrosine-based inhibitory motif (ITIM) in the cytoplasmic tail. These receptors are pivotal for regulating NK-cell activation [27,28]. NK-cell responsiveness is enhanced through a process known as “NK-cell licensing,” in which interactions between inhibitory receptors and self-HLA class I molecules prime NK cells for effective function. This interaction amplifies the NK cell’s ability to recognize and eliminate diseased cells while maintaining tolerance to healthy tissue. In contrast, NK cells lacking inhibitory receptors for HLA class I are generally hyper-responsive, but their activation depends on particularly strong activating signals. Licensed NK cells, therefore, strike a more balanced and efficient response, enabling them to counter threats effectively without causing self-reactivity. In addition to their role in licensing, iKIRs also function as inhibitory checkpoints by binding to the abundant HLA class I molecules on healthy cells, preventing licensed NK cells from attacking these cells. This balance ensures that NK cells can fight tumours and infections while sparing healthy tissue [29]. In pathological conditions, such as viral infections or cancer, diseased cells often downregulate HLA I expression (“missing-self-hypothesis” [30]) to escape recognition through CD8^+^ T cells. In turn, this triggers NK-cell activation, as now the signals of the inhibitory receptors are absent, which then activates the NK cells against the tumour cells in interaction with the HLA-independent activating receptors like NCR and NKG2D [19]. The non-HLA-specific receptors such as PD-1, SIGLEC-7, LAIR-1, TIGIT, and IRp60 contribute to regulating immune responses. All these inhibitory receptors function as immune checkpoints, modulating the anti-tumour activity of NK cells by detecting specific ligands on tumour cells, thereby enabling tumours to evade NK-cell-mediated cytotoxicity [31].

#### 1.1.3. The Balance Between Inhibition and Activation

The interplay between inhibitory and activating signals ensures the functional flexibility of NK cells, allowing them to adapt to complex microenvironments. While the inhibitory signals through KIRs maintain self-tolerance, strong activating signals, such as those mediated by NKG2D and cytokine stimulation, can override this inhibitory pathways, ensuring the effective elimination of malignant cells [32].

When activating signals predominate, NK cells can destroy target cells through granule-mediated cytotoxicity or receptor-induced apoptosis. The cytolytic effector response is primarily mediated by the release of cytotoxic granules containing perforin, which creates pores in the target cell membrane, and granzymes, which enter through these pores to trigger apoptosis. NK cells also secrete pro-inflammatory cytokines such as IFNγ and TNFα, which amplify immune responses by recruiting and activating dendritic cells, macrophages, and T cells [33]. Moreover, NK cells can employ caspase-mediated apoptosis through the expression of death receptor ligands, such as Fas ligand (FasL) and TNF-related apoptosis-inducing ligand (TRAIL), which facilitates the binding of these ligands to their respective receptors on target cells, thereby triggering programmed cell death [34,35]. Additionally, NK cells mediate ADCC by engaging the CD16 (FcγRIIIA) receptor. This receptor is capable of recognizing the Fc regions of IgG antibodies coating opsonized cells, thereby further enhancing their ability to eliminate threats. More precisely, NK cell-mediated ADCC kills target cells by releasing cytotoxic granules, activating TNF family death receptors, or secreting cytokines like IFNγ. While the release of granules like perforins and granzymes and death receptor signalling trigger apoptosis in target cells, IFNγ stimulates nearby immune cells to enhance antigen presentation and support the adaptive immune response [36].

#### 1.1.4. The Interplay Between NK Cells and Tumours

NK cells play a pivotal role in the process of cancer immunosurveillance, with their potent cytotoxic and immunomodulatory functions enabling the recognition and elimination of tumour cells. However, this interplay evolves dynamically during cancer progression. The repertoire of NK-cell receptors is directly related to their anti-tumour function. During the initial phase, stressed tumour cells express distinct ligand profiles recognized by NK cells. Over time, tumour editing occurs, selecting for NK-resistant variants. Tumours can evade NK-cell surveillance by downregulating ligands or creating an immunosuppressive TME. However, if the tumour re-expresses activating NK-cell ligands or pro-inflammatory factors, NK cells can induce tumour inflammation by producing chemokines and thereby recruiting other immune cells, converting the “cold” tumour into an immunologically “hot” state. In such cases, conventional type 1 dendritic cells (cDC1s) are recruited via chemokines to present tumour (neo-)antigens to CD8^+^ T cells in draining lymph nodes. This process, in turn, activates (neo-)antigen-specific CD8^+^ T cells, which migrate towards tumours, especially to CCL5-expressing tumours. This enhancement of tumour immunity may result in tumour regression. Conversely, the subsequent upregulation of inhibitory ligands like PD-L1 on tumour cells dampens CD8^+^ T-cell responses via PD-1-receptor engagement, contributing to tumour resistance. This cascade underlines the complex interplay between NK cells and tumours, highlighting the potential of therapeutic strategies to enhance NK-cell function and counteract tumour immune evasion mechanisms [24].

### 1.2. Solid Tumours and Immunotherapy Challenges

Solid tumours, in contrast to haematological cancers such as leukaemia and lymphoma, are broadly categorized as a malignant neoplasm arising from epithelial, glandular, and connective tissue. Those tumours are intricate and dynamic entities characterized by structural and functional heterogeneity, which drive their growth, progression, and ability to evade immune responses [37]. Solid tumours exist within a unique and complex TME, consisting of cancer cells, stromal cells, immune cells, and extracellular matrix (ECM) components. The TME is highly heterogeneous and plays a critical role in promoting tumour survival, immune evasion, and metastasis [38]. Conditions commonly associated with the TME—such as hypoxia, acidic environment, nutrient deprivation, and abnormal vasculature—further enhance the complexity of tumour progression. These factors have significant immunosuppressive effects, especially on NK cells, which are crucial for anti-tumour immunity. The TME produces various soluble factors including cytokines, lactate, and prostaglandins that have the capacity to suppress NK-cell metabolism and impair their immunological functions. Among these, the transforming growth factor-β (TGFβ) is especially significant. TGFβ downregulates NK cell-activating receptors such as NKG2D, DNAM1, and NKp30, reducing their ability to degranulate, produce cytokines, and engage in metabolic and mTOR signalling pathways, thereby diminishing their cytotoxic function [4,19,39]. Furthermore, TGFβ plays a broader immunosuppressive role by increasing the expression of CXCR3 (C-X-C motif chemokine receptor 3) and CXCR4 (C-X-C motif chemokine receptor 4) on NK cells, interfering with their egress from the bone marrow and inhibiting their maturation into functional immune cells [40]. TGFβ`s impact is not limited to NK cells; its receptors are expressed by most immune cells, allowing TGFβ to exert widespread immunosuppressive effects within the TME. Of note, high levels of TGFβ expression are associated with poor prognoses in several solid tumours, including liver cancer, lung cancer, breast cancer, pancreatic cancer, and gastric cancer [41,42,43]. Therefore, it represents a worthy target for improving immunotherapies and enhancing NK-cell function. Another major barrier to NK-cell efficacy is the physical and mechanical characteristics of solid tumours. The dense ECM and increased interstitial pressure within tumours impede the penetration and navigation of immune cells, including NK cells [44]. Stromal cells and ECM components further restrict NK-cell infiltration into the tumour core, reducing their capacity to eliminate malignant cells. This physical barrier is a significant limitation to the success of cancer immunotherapies [45], especially since NK-cell infiltration in solid tumours correlates with positive prognostic outcome [11,39,46]. Moreover, the antigenic heterogeneity displayed by solid tumours complicates their targeting by NK-cell therapies. Combined with TME suppression and poor tumour infiltration, these factors diminish the unassisted effectiveness of NK cell-based approaches [39]. Additional limitations include the short in vivo half-life of NK cells [47] and their short-term anti-cancer activity [48], both of which constrain their therapeutic potential. Addressing these challenges requires innovative strategies to optimize NK cell-based immunotherapies. Approaches such as blocking inhibitory checkpoints, boosting NK-cell activation, modifying the TME to avoid its immunosuppressive effects, or antigen-specific NK-cell engineering are critical for advancing NK-cell therapies. These interventions aim to restore NK-cell function, improve infiltration, and sustain their anti-tumour responses, offering new hope for the treatment of solid tumours.

### 1.3. Introducing Antigen Specificity by NK-Cell Engineering

#### 1.3.1. Chimeric Antigen Receptors

Chimeric antigen receptors (CARs) are engineered fusion proteins that redirect immune effector cells to tumour cells by recognizing specific antigens on their surface. In general, a typical CAR construct comprises three main components: an extracellular antigen-binding domain (usually a single-chain variable fragment (scFv)) that recognizes the desired tumour antigen; a spacer or hinge region that connects the extracellular binding region to the transmembrane domain; and an intracellular signalling domain. Originally, these CAR-constructs were used to retarget T cells to antigens expressed on tumour cells. CARs can bind directly to the target antigen presented on tumour cells without the involvement of HLA complexes, as occurs in T-cell receptor (TCR) binding. This allows for target-specific cytotoxicity [39]. (CAR)-engineered NK cells (CAR-NKs) have emerged as a highly promising approach due to their ability to specifically target tumour-associated antigens and enhance therapeutic efficacy [4]. CAR-NK cells offer several advantages over traditional CAR-T cells in cancer immune therapy. They exhibit a significantly lower risk of adverse effects such as cytokine release syndrome (CRS), neurotoxicity, and graft-versus-host disease (GvHD), making them safer for clinical use [49]. Additionally, the lack of GvHD allows the production of allogenic CAR-NK cells as “off-the-shelf” products, enhancing their accessibility and feasibility [3]. A summary of advantages of CAR-NK cells over CAR-T cells is provided in Table 1.

**Table 1 ijms-26-06290-t001:** Advantages of CAR-NK cells over CAR-T cells [49,50].

Feature	CAR-T Cells	CAR-NK Cells
Cost	-High	-Low due to off-the-shelf potential
Side effects	-GVHD ^1^ -ICANS ^2^ -CRS ^3^ -TLS ^4^	-Flu-like symptoms -Rare cases of CRS reported
Killing mechanism	-Single mechanism: CAR-mediated killing	-Dual mechanisms: CAR-mediated killing (direct) + NK receptor-dependent killing (indirect) → multiple tumour recognition mechanisms + natural ability against non-self
Source	-Mainly autologous T cells	-Variety of sources (see Table 2)
Self-recognition of normal cells	-No intrinsic self-recognition system → higher risk of on-target/off-tumour toxicity	-Self-identification of normal cells via KIRs ^5^ and NKG2A
Immune escape	-High	-Low → can kill tumour cells that downregulate HLA ^6^

^1^ GVHD: graft-versus-host disease; ^2^ ICANS: immune effector cell-associated neurotoxicity syndrome; ^3^ CRS: cytokine release syndrome; ^4^ TLS: tumour lysis syndrome; ^5^ KIR: killer cell immunoglobulin-like receptor; ^6^ HLA: human leukocyte antigen.

The first-generation CARs feature a simple structure consisting of an extracellular scFv for antigen binding and an intracellular signalling domain essential for cell activation, for instance CD3ζ (CD3 zeta chain) (Figure 3). However, the absence of co-stimulatory domains in this CAR design limited cytokine secretion, signalling efficiency, persistence, and overall anti-tumour activity. Second- and third-generation CARs address these limitations by incorporating one or two co-stimulatory signalling domains, such as 4-1BB (CD137), CD28, DAP10, DAP12, 2B4, OX40 (CD134), or FceRIg for NK cells (Figure 3). These modifications enhance cell activation, proliferation, survival, and anti-tumour efficacy. Fourth-generation CARs introduce immune modulators into the tumour microenvironment upon activation, promoting the recruitment and activation of additional immune cells to amplify the immune response against tumour cells (Figure 3). The fifth-generation CARs include shortened intracellular segments of cytokine receptors containing motifs that bind transcription factors, further improving signalling and therapeutic potential (Figure 3) [49,51,52].

Researchers are currently exploring strategies to optimize CAR-NK cells for solid tumour treatment by incorporating autocrine cytokines, modifying metabolic profiles, and blocking immune checkpoint proteins. These enhancements aim to improve NK-cell cytotoxicity, tumour infiltration, and therapeutic durability. For instance, cytokines like IL-2 and IL-15 have been shown to boost NK-cell proliferation [53]. Additional approaches involve protecting CAR-NKs from inhibitory signals in the TME and equipping them with chemokine receptors to facilitate better tumour penetration [19].

#### 1.3.2. NK-Cell Sources and CAR Transfer Methods

Despite their promise, there are several challenges in CAR-NK-cell production, particularly in selecting the source of NK cells. Potential sources include the immortalized NK92 cell line, peripheral blood (PB), umbilical cord blood (UCB), haematopoietic stem cells (HSCs), and induced pluripotent stem cells (iPSCs). Each source has distinct advantages and limitations (Table 2). An ideal NK-cell source should allow for the generation of a high number of NK cells, while minimizing manufacturing complexity, costs, and product heterogeneity [4].

The gene transduction methods used to generate CAR-NK cells include lentiviral and retroviral vectors, electroporation, liposomes, and DNA transposons [54]. While viral transduction has been widely used, it raises safety and regulatory concerns. Retroviral vectors, for instance, can integrate into the genome, posing risks such as cellular transformation and cancer development. Lentiviral transduction has a lower genotoxicity but requires several rounds of transduction due to low efficiency, making the process expensive and labour-intensive. As a result, non-viral approaches are gaining interest due to their improved safety profiles. Electroporation and liposome-based transfection, for example, can efficiently introduce exogenous genes into NK cells, resulting in rapid gene expression, with lower apoptosis rates, minimal immunogenicity, and reduced production costs. However, because these methods deliver RNA/DNA without integrating it into the genome, gene expression remains transient, necessitating alternative approaches for stable modification [19,55,56].

**Table 2 ijms-26-06290-t002:** Comparative overview of NK-cell sources [3,39,57].

NK Cell Sources	Advantages	Limitations
NK92 cells	-Easy to expand. -Easily genetically manipulated; allows introduction of exogenous genes through electroporation; no viral vectors needed. -Only NK-cell line applied in clinical trials; demonstrates controllable adverse effects. -Multiple doses available.	-Require irradiation before use to inhibit in vivo proliferation, limiting long-term persistence. -Lack of CD16 receptor-mediated ADCC ^1^ effect.
PB ^2^ NK cells	-Mature NK cells ready for immediate use without differentiation. -Easy in vivo expansion. -Well-documented clinical safety and efficacy.	-Endogenous NK-cell function may be impaired by disease or treatment. -Requires T cell removal to prevent GvHD ^3^ in allogenic settings. -Limited to a single dose per donor. -Genetic modifications are challenging. -Lower percentage of NK cells in PB.
UCB ^4^ NK cells	-High proliferation efficiency. -Robust bone marrow homing ability. -Higher percentage of NK cells. -Relatively stable NK-cell source. -Ability to cryopreserve them.	-Not fully differentiated. -Low expression of NK-cell receptors. -Limited cell inhibition ability. -Potential risk of tumourigenesis in allogenic transplantations.
HSC ^5^ NK cells	-Potential for unlimited numbers. -Amenable to genetic manipulation. -Suitable for “off-the-shelf” therapy.	-Possibility of triggering unexpected immune responses due to potential immunogenicity. -Long expansion time.
iPSC ^6^ NK cells	-Efficiently clone, expand, and differentiate in vitro. -Uniform cell products. -Multiple doses available from single source. -Suitable for “off-the-shelf” therapy.	-Complicated production process. -Low expression of endogenous CD16. Limiting ADCC. -Risk of malignant transformation. -Potential immunogenicity concerns.

^1^ ADCC: antibody-dependent cellular cytotoxicity; ^2^ PB: peripheral blood; ^3^ GvHD: graft-versus-host disease; ^4^ UCB: umbilical cord blood; ^5^ HSC: haematopoietic stem cells; ^6^ iPSC: induced pluripotent stem cells.

### 1.4. Key Advancements in CAR-NK Cell Therapy

CAR-NK therapy offers several advantages over CAR-T therapy. A major advantage of CAR-NK cells is their ability to function in allogenic settings without requiring strict HLA-matching, as NK cells are not restricted by self-HLA recognition. This greatly reduces the risk of severe side effects such as GvHD, immune effector cell-associated neurotoxicity syndrome (ICANS), and CRS. Although flu-like symptoms are relatively common and rare cases of CRS have been reported after CAR-NK therapy, these side effects are generally manageable and less severe compared to CAR-T therapy [58]. Another advantage of the possibility to use CAR-NK cells in allogenic settings is the possibility of “off-the-shelf” production of larger batches for more than one patient, which significantly reduces manufacturing complexity and associated costs. Since CAR-NK cells can be derived from various sources, like UBC or IPCs, it allows the possibility of a ready-to-use product. In contrast, CAR-T cells, which require patient-specific manufacturing, are cost-intensive and time-consuming. This scalability is a key factor in broadening the accessibility of CAR-NK therapies. Moreover, NK cells possess dual killing mechanisms, as described above. This dual mechanism enhances their versatility and efficacy compared to CAR-T cells, which rely solely on antigen-specific CAR-signalling. Furthermore, CAR-NK cells have immunomodulatory functions, which can regulate the function of B cells and cytotoxic lymphoid cells [59].

Despite these strengths, CAR-NK therapy still presents challenges. A major restriction is the limited persistence of NK cells in vivo in solid tumours due to their natural short life span, rapid turnover, and the harsh TME. These conditions promote apoptosis and dysfunction while persistent activation can lead to exhaustion, reducing cytotoxicity [60]. One approach to counteract these limitations is the genetic modification of CAR-NK cells to express cytokines that support growth and survival. For instance, IL-15 plays a crucial role in NK-cell development, differentiation, and function [61]. It is essential for NK-cell homeostasis, regulating their survival and proliferation through maintenance of the anti-apoptotic factor Bcl-2 (B-cell lymphoma 2) [62]. Therefore, it is a reasonable approach to use IL-15 to improve CAR-NK cell function. Recent studies have explored that equipping CAR-NK cells with IL-15 can make additional cytokine supplementation obsolete [63,64,65]. For instance, van Eyden et al. demonstrated the successful incorporation of an IL-15 targeting cassette into a CD70-targeting CAR, resulting in enhanced persistence and an increased activation status in both in vitro and in vivo models of pancreatic cancer [66]. This approach has so far demonstrated the most significant clinical outcome, with studies reporting persistence for over one year following infusion in lymphoid tumours [67] and over 90 days after the last infusion in solid tumours [68]. Similarly to the incorporation of IL-15, He et al. demonstrated that CAR-NK cells co-expressing IL-21 exhibited higher expansion and secretion of IFNγ and TNFα in response to CD19^+^ lymphoma [69]. Also for CAR-NK cells, it could be demonstrated that the co-expression of IL-21 enhanced anti-tumour activity [70]. Another promising approach involves the harnessing of memory NK cells. This is based on evidence suggesting that NK cells show features of adoptive memory [71]. Analogous to memory B and T cells, memory NK cells can generate an enhanced response upon re-exposure to a previously encountered stimulus and are sustained by a long-lived, self-renewing population [72]. For clinical usage, cytokine-induced memory-like (CIML) NK cells have shown encouraging results and are a promising strategy for CAR-NK cell therapy [73,74].

Besides the limited persistence, the reduced ability of CAR-NK cells to infiltrate and traffic within the TME is a major challenge, hindering their effectiveness against solid tumours. There are several strategies to improve CAR NK-cell trafficking and infiltration. One approach involves the manipulation of chemokine receptors and their respective ligands, since chemokine receptors play an essential role in the migration of lymphocytes and are critical in driving tumour infiltration of NK cells via interactions between soluble chemokines and their receptors. This is of particular importance given the ability of solid tumours to alter the chemokine receptor/ligand axes, thereby impairing the trafficking of NK cells. In the context of NK cells, important chemokine receptors include CCR2, CCR5, CCR7, CXCR1, CXCR2, CXCR3, and CX3CR1 [75]. For instance, Yoon et al. demonstrated that anti-mesothelin CAR-NKs with incorporated CXCR2 exhibited enhanced tumour killing and strong infiltration of tumour sites in pancreatic 2D cancer cell co-culture and 3D tumour-mimetic organoid models [76,77]. Yang Ng et al. followed a similar but slightly different approach. They electroporated NK cells with two mRNA constructs encoding the chemokine receptor CXCR1 and a CAR construct targeting tumour-associated NKG2D ligands. The co-expression of CXCR1 and a CAR resulted in enhanced migration towards the pro-inflammatory cytokine IL-8 gradient, leading to increased infiltration and anti-tumour response in mice carrying established peritoneal ovarian cancer xenografts [78]. The incorporation of the CCR4 chemokine receptor in a CAR-construct was successful as well [79]. Other strategies to improve the infiltration and trafficking of CAR-NK cells are nano-enzyme-armed NK cells [80], targeting signalling pathways [81], and combination therapies [82].

A further issue is the limited expansion of CAR-NK cells in vivo after transplantation, due to tumour heterogeneity and a lack of cytokines, e.g., IL2 and IL15. This means the number of NK cells from a single donor is often insufficient for treatment, making the expansion of primary NK cells in vivo an important task [83]. One approach involves the supplementation of cytokines to enhance NK-cell activation. For instance, IL-2 stimulates NK-cell activity [84], further IL-15 [85] and IL-21 [86] have shown potential in promoting NK-cell activation, proliferation, and expansion. IL-18 is another cytokine that is currently under consideration [87]. Moreover, as previously noted, incorporating those cytokines directly into the CAR construct offers a promising strategy to further boost NK-cell function.

However, large-scale production and clinical translation remain significant challenges. At present, no FDA-approved drugs for CAR-NK therapies exist, although several are being evaluated in clinical trials [49]. With a growing number of clinical trials, their relevance in cancer immunotherapy becomes increasingly important.

## 2. Potential Therapeutic Implications

### 2.1. Preclinical Studies

The functionality of CAR-NK cells is currently tested in several preclinical studies for different types of solid tumours, for instance in breast cancer. Triple-negative breast cancer (TNBC) is a particularly aggressive subtype of breast cancer characterized by the absence of estrogen receptors, progesterone receptors, and overexpression of human epidermal growth factor receptor 2 (HER2) limiting treatment options. In a recent study, Yang et al. demonstrated the potential of CAR-NK cells for TNBC patients by using mesothelin (MSLN)-targeted CAR-NK cells derived from induced pluripotent stem cells. These cells effectively eliminated TNBC cells in vitro, in vivo, and ex vivo, highlighting mesothelin as a promising target [88]. Furthermore, Liu et al. showed that CAR-NK cells targeting human epidermal growth factor 1 (HER1) holds promise for TNBC patients [89]. Further, Raftery et al. identified CD44v6, an adhesion molecule expressed in solid tumours that is involved in tumourigenesis and metastases, as a viable target. The subsequent generation of primary CAR-NK cells directed against CD44v6, incorporating an IL-15 superagonist and checkpoint inhibitor molecules, exhibited effective anti-tumour activity against 3D spheroid models and resistance to the TME [90]. Hu et al. further demonstrated the efficacy of tissue factor (TF)-targeting NK92MI CAR-NK cells co-expressing CD16, in killing TNBC cells, with L-ICON ADCC in vitro [91]. Moreover, the potential of EGFR (epidermal growth factor receptor)-targeting CAR-NK cells for TNBC has been demonstrated [92,93]. In the context of HER-2^+^ breast cancer, HER-2-targeting NK92 CAR-NK cells exhibit enhanced cytotoxicity [64].

CAR-NK therapy is also being explored in glioblastoma (GBM). A number of studies have reported encouraging preclinical results for CAR-NK cells targeting several antigens, such as HER2 [94,95,96], EGFR [97,98,99], and B7-H3 (CD276) [100,101,102,103]. Of note, Tachi et al. demonstrated that B7H3-targeting, CB-derived CAR-NK cells suppressed glioblastoma tumour growth and extend survival in xenograft models using patient-derived GBM cells, which accurately reflected characteristics of patient tumours [100]. In a related study, Lima et al. proposed a novel approach to target glioblastoma stem cells (GSCs), which are resistant to conventional treatments and possess tumour-initiating properties, with third-generation CAR-NK cells. The GSC-targeting NK92 CAR-NK cells showed convincing cytotoxicity in GMB cell lines [104].

In the context of lung cancer, CAR-NK cells have demonstrated substantial therapeutic potential [101,105]. For instance, Zhang et al. provided a novel option for treating lung cancer using NKG2D-IL-21 CAR-NK-cell therapy with NKs from the NK92 cell line. The NKG2D-IL-21 CAR-NK cells showed enhanced anti-tumour activity against lung cancer cells and suppressed tumour growth in vitro and in vivo. Furthermore, the study noted enhanced proliferation, alongside suppressed apoptosis and exhaustion of the cells [70]. In a further study, Liu et al. engineered delta-like ligand 3 (DLL3)-specific CAR-NK92 cells for small-cell lung cancer, achieving significant in vitro cytotoxicity and elevated levels of cytokine production. Moreover, the DLL3-CAR-NK92 cells were able to infiltrate subcutaneous tumour models and exhibited potent anti-tumour activity within a safe therapeutic range [106]. For non-small lung cancer, Zolov et al. developed a new CAR construct targeting cell adhesion molecule 1 (CADM1), which showed increased cytotoxicity against CADM1-expressing lung cancer cells in comparison to non-transduced NK92 cells [107]. Using a different approach, Chambers et al. designed PB-CAR-NK cells targeting the CD73 adenosine axis through blocking the enzymatic activity of CD73, resulting in an impaired adenosinergic metabolism and induction of tumour stasis. This promoted NK-cell infiltration into CD73 tumours and enhanced intratumoural infiltration [108].

Especially for pancreatic cancer, which is highly aggressive and for which effective treatments are limited, CAR-NK cells offer new potential for better therapy. Numerous preclinical studies supported this hypothesis: these include anti-CD70 CAR-NK92 cells [66], anti-mesothelin-CAR-NK92 cells [109], CXCR2-augmented PB-CAR-NK cells [76], and prostate stem cell antigen (PSCA)-directed UCB-CAR-NK cells secreting soluble IL-15 [68]. Similarly, in liver cancer, preclinical studies support the therapeutic potential of CAR-NK cells [110,111,112].

Recent studies also investigated the suitability of CAR-NK cells in the treatment of gastric cancers. For instance, MSLN-CAR-NK cells were found to be capable of specifically eliminating MSLN^+^ gastric cancer cells in vitro, while sparing MSLN-negative cells. Furthermore, the CARs demonstrated notable efficacy in eradicating cancer cells and infiltrating both subcutaneous and intraperitoneal tumour models, as well as patient-derived xenograft tumour models [113]. A study by Ren et al. proposed an interesting new approach. They designed a novel chimeric cytokine receptor TRII/21 R, which consists of extracellular domains of the TGFβ receptor II and transmembrane and intracellular domains of the IL-21 receptor. It has the capacity to convert immunosuppressive signals from TGFβ in the TME into an NK-cell activation signal through the IL-21R-STAT3 pathway. Subsequent construction of NKG2D-CAR-NK cells expressing TRII/21 R resulted in strong anti-tumour activity against cancer cells in both in vitro and in vivo models. In addition, the co-expression of TRII/21 R in CAR-NK92 cells leads to increased cytotoxicity, heightened proliferation and survival capabilities, and reduced expression of exhaustion markers in xenograft mouse models [114]. Furthermore, the preclinical investigation of CAR-NK92 cells in colorectal cancers has yielded encouraging results as well [79,115,116,117]. Exploration is ongoing for their potential in renal [118] and ovarian cancers [119,120,121], further demonstrating their broad applicability for solid tumour immunotherapy.

### 2.2. Clinical Studies

Several clinical trials are currently assessing the safety and efficacy of tumour-targeting CAR-NK cells for the treatment of solid tumours. Table 3 summarizes both ongoing and completed clinical trials investigating the potential of CAR-NK-cell therapy. These trials are examining critical aspects of CAR-NK therapy, including response rates, durability, and potential adverse effects. The first clinical trial of CAR-NK therapy for solid tumours used MUC1 (Mucin 1) CAR-NK cells to target malignant solid tumours, including glioblastoma, pancreatic, colorectal, breast, and ovarian cancers (NCT02839954) and was quite promising, with seven out of eight patients achieving stable disease without serious side effects [122].

Currently, the application of CAR-NK cell therapy is investigated in a broad range of solid tumours, including pancreatic cancer (NCT03941457, NCT06464965, NCT05922930, NCT06816823, NCT06478459), ovarian cancer (NCT05922930, NCT05856643, NCT03692637, NCT05410717), hepatocellular carcinoma (NCT06652243), prostate cancer (NCT03692663), colorectal cancer (NCT05213195), and gastric cancer (NCT06464965). Preliminary findings suggest that CAR-NK cells exhibit anti-tumour activity with manageable toxicity. For instance, preliminary data from a phase 1 trial with anti-HER2 CAR-NK cells (NCT04319757) showed no CRS, GvHD, neurotoxicity, and no dose-limiting cytotoxicity [123]. In addition, data from a phase 1 trial with NKG2DL CAR-NK cells in three patients with colorectal cancer (NCT03415100) demonstrated no serious adverse effects and no dose-limiting side effects [115], with only grade 1 CRS observed. In the first two patients treated with an intraperitoneal infusion of CAR-NK cells, a noticeable reduction in the number of tumour cells in the ascites was observed, indicating a direct therapeutic effect against metastases. The third patient, who has developed liver metastases, showed rapid tumour regression at the injection site. The CARs demonstrated effective recognition and destruction of tumour cells, as evidenced by a significant loss of NKG2D ligands on the biopsy samples from injected sites, along with necrosis of tumour cells, suggesting that the CAR-NK cells effectively eliminated tumour cells. However, in rare cases, as evidenced by a clinical phase 1 study involving CCCR-NK92 cells against non-small cell lung cancer, one patient experienced CRS. Despite the reduced risk of toxic side effects in CAR-NK cell therapy, it is crucial to employ a low dosage, reduce the tumour burden prior to CAR-NK cell treatment, and monitor the patient’s stability (NCT03656705) [124]. A phase 1/2 clinical trial (NCT05410717) evaluates Claudin-6-targeting CAR-NK cells for the treatment of stage IV ovarian cancer, refractory testis cancer, and recurrent endometrial cancer. In order to enhance their killing capability, the trial uses next-generation CAR-NK cells which are engineered to express and secrete IL-7/CCL19 and/or scFvs against PD-1/CTLA-4/Lag-3 [125].

However, despite promising results, further clinical trials are required to optimize CAR-NK cell therapy for clinical application, especially since there are currently no phase 3 and 4 trials.

### 2.3. Combination Therapies in Preclinical and Clinical Settings

A potential solution for the challenges already described above is the combination of CAR-NK cells with other immunotherapies, radiotherapy, or chemotherapy, which could enhance anti-tumour activity and improve therapeutic outcomes [82] (summarized in Table 4).

#### 2.3.1. CAR-NK Cells Combined with Immune Checkpoint Inhibitors

One promising approach involves the combination of CAR-NK-cell therapy with immune checkpoint inhibitors (ICIs). This strategy may prevent tumour-induced shifts in NK-cell receptor–ligand interactions, a mechanism through which tumours evade immune surveillance by increasing inhibitory receptor expression while downregulating activating receptors [136]. ICIs targeting programmed cell death protein 1/programmed cell death protein 1 ligand (PD-1/PD-L1) and cytotoxic T-lymphocyte-associated protein 4 (CTLA-4) in T cells have revolutionized cancer treatment and have shown impressive clinical results in solid tumours such as melanoma [137]. These therapies, especially PD-1 and PD-L1 inhibitors, aim to counteract immunosuppressive pathways within the TME. In light of recent reports indicating the potential of combining CAR-T cells with checkpoint inhibitors, particularly the blocking of PD-1 receptors on CAR-T cells [138,139,140], researchers have expanded towards exploring immune checkpoints on NK cells [141]. According to recent research, PD-1 expression is also upregulated on NK cells within the TME, where its upregulation suppresses NK-cell degranulation and cytotoxicity [141,142,143,144]. Blocking PD-1 or PD-L1 has been demonstrated to restore NK-cell function, promote proliferation, and enhance cytotoxic activity against PD-L1^+^ tumours [145,146]. In the context of multiple myeloma, NK cells with high PD-1 expression regained anti-tumour activity after PD-1 antibody treatment [142]. Furthermore, CAR-NK92 cells show increased PD-1 expression following continuous activation [147]. Consequently, the combination of PD-1 antibodies with CAR-NK-cell therapy emerges as a potential strategy to reverse tumour escape pathways, reverse CAR-NK-cell exhaustion, and enhance their efficacy [148]. A preclinical study combining CAR-NK cells, targeting HER2, and anti-PD-1 checkpoint inhibitors demonstrated a synergistic anti-tumour effect in a mouse glioblastoma model. More precisely, the study showed that the CAR-NK cells (NK92/5.28.z cells) were significantly improved in their therapeutic efficacy when combined with an anti-PD-1 antibody, resulting in the successful treatment of advanced tumours that were resistant to NK92/5.28.z monotherapy. Moreover, the combination therapy induced a cytotoxic rather than immunosuppressive TME, and activated the immune system [149]. The success of these findings led to clinical investigations, such as the CAR2BRAIN study (NCT03383978), which combines NK92/5.28.z cells with intravenous Ezabenlimab in patients with recurrent HER2-positive glioblastoma. In this clinical trial, no dose-limiting toxicities, CRS, or neurotoxicity were observed. Of nine patients treated with NK92/5.28.z cells after relapse after surgery, five achieved stable disease lasting 7–37 weeks, while four had progressive disease. Pseudoprogression at injection sites in two patients suggested immune activation. Median progression-free survival and overall survival were 7 and 31 weeks, respectively. Higher levels of CD8^+^ T-cell infiltration in tumours were associated with delayed disease progression [94]. Although this therapy seems promising, the group of Liu et al. focused on a slightly different approach. They proved that CAR-NK cells harbouring anti-PD-L1 properties in combination with the checkpoint inhibitor Nivolumab led to tumour regression in a nasopharyngeal cancer (NPC) mouse model [127]. The anti-PD-L1 CAR-NK, combined with anti-PD-1 and IL-15 super antagonist N-803, enhanced NK-cell cytotoxicity in a mouse oral cancer syngeneic model, while reducing T-cell exhaustion and stimulating T-cell responses through IL-15-induced IFNγ release. Further, a clinical trial (NCT04847466), which is currently in the recruiting state, investigates the combination of Pembrolizumab, blocking the PD-1 receptor, irradiated PD-L1 CAR-NK cells, and N-803 to target refractory gastric and head and neck malignancies. Besides PD-1, other inhibitory receptors such as TIGIT (T cell immunoreceptor with Ig and ITIM domains), LAG-3 (lymphocyte-activation gene 3), and TIM-3 (T-cell immunoglobulin and mucin domain 3) [150,151] are also potential targets for combination therapies aimed at enhancing NK cell-mediated tumour destruction. [152,153].

#### 2.3.2. CAR-NK Cells Combined with Chemo- or Radiotherapy

Moreover, combining CAR-NK-cell therapy with chemotherapy or radiotherapy may further enhance their efficacy. Numerous studies demonstrated that chemoradiotherapy can reduce the proliferation of cancer cells and plays a role in regulating the expression of NK-cell receptors and ligands. By targeting the immunosuppressive TME and by enhancing the recruitment of CAR-NK cells, this approach has the potential to improve the efficacy of immune cells in eliminating cancer cells [19]. Recent studies confirmed that treating ovarian cancer with cisplatin followed by CAR-NK cells effectively killed cancer stem cells [128,129]. This finding implies that CAR-NK cells remain toxic in the presence of cisplatin, thereby enhancing their anti-tumour activity. Moreover, CAR-NK cells are being explored as drug delivery systems, where they transport chemotherapeutic agents directly to tumour sites, reducing systemic toxicity while increasing local efficacy [154,155,156]. For instance, Siegler et al. demonstrated that CAR-NK cells with cell surface-bound multilamellar liposomal vesicles loaded with the chemotherapeutic drug PTX (paclitaxel) enhance the delivery of PTX to the tumour site, slowing tumour growth and increasing intratumoural PTX concentrations more effectively in vivo and ex vivo, without causing toxicity to the carrier NK cells. The authors even suggest that their CAR-NK-mediated drug delivery system could be expanded to include the delivery of other anti-cancer treatments such as immunomodulators and small molecules that affect the TME [154,155]. When it comes to the combination of CAR-NK cells with radiotherapy, this approach holds promise for improving NK-cell trafficking, infiltration, and tumour recognition [157]. The group of Lin et al. provided evidence that irradiation could efficiently enhance the anti-tumour effect of CAR-NK cells targeting Glypican-3 (GPC3) on hepatocellular carcinoma cells in vivo. Of note, it only worked with high-dose irradiation (8 Gy) [131]. Nevertheless, more studies are required to determine the optimal dose, duration, and sequence of this combination of therapies and the potential of these approaches in other types of solid tumours, as well as to avoid the immunosuppression mediated by radiotherapy.

#### 2.3.3. CAR-NK Cells Combined with STING Agonists

Another emerging strategy involves the combination of CAR-NK cells with stimulator of interferon genes (STING) agonists, which activate the STING signalling pathway in NK cells. This activates NK cells by upregulating the expression of activating receptors and downregulating inhibitory receptors, thereby amplifying the anti-tumour activity of NK cells. This novel strategy for tumour immunotherapy has been demonstrated in pre-clinical studies [158,159]. For instance, a combination of CAR-NK cells targeting mesothelin and the STING agonist cGAMP (cyclic GMP-AMP) showed improved anti-tumour efficacy in a mouse model of pancreatic cancer, indicated by the inhibition of tumour growth and the prolongation of survival, in comparison to either treatment alone [109]. Another study demonstrated that the combination of STING agonists and CAR-NK cells is also successful in patient-derived organotypic tumour spheroids [132]. While the combination of CAR-NK cells and STING agonists might be a promising approach to fight cancer, further clinical studies are required to validate these results in humans.

#### 2.3.4. CAR-NK Cells Combined with Enzyme Inhibitors

A distinct methodology involves the usage of enzyme inhibitors in combination with CAR-NK cells, such as tyrosine kinase inhibitors (TKIs) and proteasome inhibitors (PIs). In particular, TKIs have emerged as a promising class of targeted cancer therapies for solid tumours since it was demonstrated that they possess direct anti-tumour activity, are able to regulate the TME, and promote anti-tumour immunity [160]. Examples of TKIs used for solid tumours include Imatinib, Dasatinib, Gefitinib, Erlotinib, Sunitinib, Cabozantinib, Sorafenib, and Regorfenib [117,161,162]. A notable finding is the synergistic effect observed in a human colon cancer model when combining Regorafenib with CAR-NK cells that target the epithelial cell adhesion molecule (EpCAM). This combination significantly enhances anti-tumour efficacy [117]. In addition, the Src kinase inhibitor Dasatinib has the potential to further enhance the anti-tumour activity of CAR-NK cells in a CAR-specific manner [163]. Further, it was demonstrated that the TKI Cabozantinib improves renal cell carcinoma-specific cytotoxicity by enhancing the expression of EGFR and reducing PD-L1 expression in renal cell carcinoma, working in a synergistic way with CAR-NK92 cells and improving their cytotoxicity both in vitro and in vivo [164]. In contrast to TKIs, protease inhibitors have so far only been successful in combination with CAR-NK cells in haematological cancers. In general, it was demonstrated that PIs inhibit tumours directly or indirectly by amplifying the function of NK cells or amplifying the sensitivity of tumour cells to killing [165,166]. In the context of acute myeloid leukaemia and multiple myeloma, the combination of PIs, in particular Bortezomib, with CAR-NK cells has been shown to result in synergistic effects [167,168]. Consequently, these studies imply that combining PIs with CAR-NK cells in solid tumours is also a plausible therapeutic strategy, with the potential to enhance anti-tumour efficacy through increased tumour cell susceptibility to immune-mediated killing.

#### 2.3.5. CAR-NK Cells Combined with Oncolytic Viruses

Recently, oncolytic viruses (OVs) have gained attention as a treatment approach for cancer, especially for patients resistant to traditional therapeutic interventions. OVs, whether natural or genetically modified, selectively infect and lyse tumour cells while preserving normal cells. This process stimulates local and systemic immune responses, thereby enhancing the infiltration of immune cells into the TME and, in turn, boosting anti-tumour immunity [169,170,171]. A study demonstrated that the combination of an oncolytic virus expressing IL-15/IL-15Rα (OV-IL-15C) and off-the-shelf EGFR-CAR-NK cells had a synergistic effect. This combination resulted in an increased suppression of tumour growth and an enhancement in survival outcomes when compared with the use of monotherapy. The observed benefits were associated with increased infiltration and activation of both NK and CD8^+^ T cells, as well as the elevated persistence of the EGFR-CAR-NK cells. The interplay of these effects transforms a “cold” TME into a “hot” TME with increased immune cells, providing a rationale to combine OV, or specifically OV-IL-15C, with EGFR-CAR-NK cells, for targeting heterogeneous glioblastoma. This combination has the potential to address two significant challenges in the treatment of solid tumours: the immunosuppressive TME and the difficulty of tumour infiltration, especially for CAR-NK cells [99]. Another study confirmed that the combinational therapy of EGFR-CAR-NK cells and oncolytic herpes simplex virus-1 could prolong the survival time of mice bearing breast cancer brain metastasis [93]. An alternative approach involves the usage of CAR-NK cells that target Avsialidase, a membrane-bound sialidase, in combination with an Avsialidase-armed oncolytic vaccinia virus for the treatment of solid tumours [172].

#### 2.3.6. CAR-NK Cells Combined with CAR-T Cells or CAR-Macrophages

It has been hypothesized that NK cells can enhance CAR-T-cell activation, migration, and fitness, thereby leading to improved anti-tumour activity, reduced exhaustion, and senescence in T cells. This suggests a logical and interesting approach to combining CAR-NK cells with CAR-T cells. This approach has the potential to leverage the benefits of both cell types, thereby enhancing the efficacy of tumour treatment [173]. Male et al. proposed an interesting CAR-NK/CAR-T cell combination therapy model. This model involves chemokine-secreting CAR-NK cells, which recognize and kill tumour cells directly. These cells also release IL-8, CCL3, and CCL5, recruiting CAR-T cells to enhance their efficacy. The CAR-T cells then secrete PD-1 blocking antibodies, thereby preventing PD-L1 mediated immune evasion and inducing cancer cell death through granzyme and perforin release, which, in turn, activates CAR-NK cells [50]. Additionally, studies showed that combining CAR-NK and CAR-T cells results in rapid and persistent tumour killing, with CAR-NK cells providing immediate cytotoxicity and CAR-T cells offering long-term persistence [174]. However, the potential side effects of combining CAR-NK cells with CAR-T cells are still unknown; thus, more research is necessary.

An alternative to this approach could be the combination of CAR-NK cells with CAR-macrophages (CAR-M). CAR-Ms have certain advantages over CAR-T and CAR-NK cells in solid tumours, including efficient infiltration of the TME, remodelling the ECM via matrix metalloproteases (MMPs) for improved tumour penetration, and the capacity to kill tumour cells through phagocytosis, ROS/iNOS release, and TLR activation. Furthermore, they boost anti-tumour immunity by releasing IL-12 to activate NK cells and act as antigen-presenting cells, thereby promoting adaptive immune responses [175,176]. Considering these advantages, it could be effective to combine CAR-M with CAR-NK cells to enhance anti-tumour efficacy. This could look like the following: Initially, IFNγ-secreting CAR-Ms would recognize and phagocyte the tumour, with continuous IFNγ secretion inducing the recruitment of chemokine-expressing CAR-NK cells. Moreover, CAR-M would secrete IL-1, IL-2, and IL-15, which would upregulate CAR-NK cells and enhance their cytotoxicity against tumour cells. Additionally, the activated CAR-NK cells secrete IFNγ and TNFα, which in turn stimulates endogenous cytotoxic T cells [50].

#### 2.3.7. CAR-NK Cells Combined with Photothermal Therapy

Another approach is to combine CAR-NK cells with photothermal therapy (PTT). The rationale behind this combination is that mild hyperthermia of a tumour can reduce its dense structure and interstitial fluid pressure, increase blood flow, release antigens, and recruit endogenous immune cells. This, in turn, could enhance the ability of CAR-NK cells to kill the tumour and reduce the challenges that TME presents for CAR-NK cells. Preclinical research demonstrated that the combination of CAR-T cells and photothermal therapy improved anti-tumour activity in animal models, without inducing systemic toxicity [177,178]. A similar approach for CAR-NK cells showed promising results. Utilizing a temperature-feedback nanoplatform for NIR-II penta-modal imaging-guided PTT combined with CAR-NK-cell immunotherapy has shown effective outcomes in treating lung cancer [135]. Overall, it is a promising and realistic strategy to combine CAR-NK cell therapy with existing or other new therapies for effectively treating solid tumours. Therefore, more clinical studies are necessary to investigate the interplay between different therapies and to avoid side effects in human patients.

## 3. Open Questions and Emerging Areas

### 3.1. Tumour Microenvironment Challenges

A suppressive TME presents a major obstacle to the effective therapeutic treatment of solid tumours. In particular, hypoxia and immunosuppressive cytokines make their effective treatment difficult. Therefore, new engineering strategies involving CARs and the search for new targets are important.

#### 3.1.1. Hypoxia in the TME

A suppressive TME presents a major obstacle to the effective therapeutic treatment of solid tumours. Hypoxia especially challenges CAR-NK cells [179]. Hypoxia is a pathological process that is common in the late stages of solid tumours and is characterized by abnormal changes in the metabolism, function, and morphological structure of the tissue. This is either caused by insufficient oxygen supply or disorders in oxygen utilization within the tissue. The consequences of a hypoxic TME can be abnormal angiogenesis, reprogramed energy metabolism, immune evasion, activated invasion and metastasis, and genomic instability [180]. Hypoxia can significantly impact NK-cell function and phenotype, which may have implications for the use of CAR-NK cells. For instance, studies have shown that under hypoxic conditions, phosphorylation levels of extracellular signal-regulated kinase (ERK) and signal transducer and activator of transcription 3 (STAT3) decrease in a protein tyrosine phosphatase 1 (SHP-1)-dependent way, which impairs the cytotoxicity of NK cells [181]. Furthermore, a study demonstrated that the degradation of granzyme B, caused by hypoxia, contributes to the impairment of NK-cell functionality [182]. Hypoxia also leads to decreased expression and function of important activating NK-cell receptors, including NKp30, NKp44, NKp46, and NKG2D. This altered phenotype is associated with an impaired capacity to target and eliminate tumour cells. Of note, the ability of to kill target cells via ADCC is not significantly impaired [183]. In addition, the cytokine production of NK cells is hindered within a hypoxic TME. Lactate, which is increased under such conditions, inhibits the upregulation of nuclear factor of activated T cells (NFAT), consequently resulting in diminished IFNγ production. A comprehensive understanding of the mechanisms by which a hypoxic/suppressive TME impairs NK-cell function is important for the adaption of CAR-NK cells to hypoxic conditions and the improvement of their functionality. For instance, Liu et al. genetically engineered HER-1-overexpressing TNBC-targeting cells with catalase to provide them with tolerance against hypoxia and oxidative stress inside TNBC tumours [89]. The catalase is expected to decompose the hydrogen peroxide, a reactive oxygen species inside tumours, into oxygen. Furthermore, they used alginate hydrogel for the intratumoural fixation of CAR-NK cells, which resulted in increased anti-tumour activity in the study. Another approach investigated by the group of Duan et al. involved the use of biodegradable manganese dioxide nanoparticles, particularly MnOX nanoenzymes, which can produce oxygen by catalysing hydrogen peroxide to O_2_ within solid tumours, thereby recovering the oxygen levels in the TME. The combination of these MnOX nanoenzymes with CAR-NK cells in mice bearing solid tumours has been shown to enhance infiltration and anti-tumour activity [80].

To date, there are only a limited number of studies that have directly addressed the potential for adapting CAR-NK cells to hypoxia within the TME of solid tumours. However, there are already several studies examining how NK cell-based therapies can be adapted, and it is conceivable that similar strategies can be applied to CAR-NK cells. One possibility that has been successfully tested by several research groups is to culture NK cells under moderate hypoxic conditions. This approach was shown to enhance their anti-tumour activity in immunosuppressive TMEs in solid tumours [184,185]. In contrast, the group of Solocinski et al. pursued a different strategy, utilizing high-affinity NK (haNK) cells. These cells have been equipped with a high-affinity CD16 receptor and internal IL-2, resulting in enhanced ADCC and activation. They demonstrated that haNKs maintained their killing capacity under hypoxic conditions, whereby the IL-2 produced by the haNKs is hypothesized to drive the maintenance of the killing capacity. Furthermore, they discovered that active STAT3 could be associated with reduced NK-cell function, which may lead to a further strategy of adapting CAR-NK cells to the TME of solid tumours [186]. However, further research is necessary, to find a way to adapt CAR-NK cells to hypoxia in the TME of solid tumours.

#### 3.1.2. Immunosuppressive Cytokines in the TME

Besides hypoxia, the presence of immunosuppressive cytokines in the TME represents another significant hindrance that must be overcome. In particular, TGFβ plays a significant role in this regard. It inhibits various aspects of NK-cell anti-tumour function, such as metabolism, cytokine secretion, degranulation, and mTOR signalling, thereby diminishing their cytotoxic activity [4,19,39]. Consequently, the inhibition of TGFβ or the deactivation of the TGFβ receptor on NK or CAR-NK cells has the potential to counteract TGFβ-mediated inhibition and enhance NK anti-cancer activity [187,188,189,190,191,192]. Building on this, Weathers et al. engineered NK cells with a genetically depleted TGFβ receptor 2 and the endogenous glucocorticoid receptor nuclear receptor subfamily 3 group C member (NR3C1), which is currently being tested in a phase I clinical trial (NCT04991870). Another clinical study (NCT05400122) aims to enhance the activity of NK cells by administering them alongside IL-2 while simultaneously inhibiting the TGFβ signalling pathway using the TGFβ receptor 1 inhibitor Vactosertib [193]. If these studies prove successful, they could be implemented in CAR-NK cells to enhance anti-tumour efficacy. Oh et al. have explored alternative strategies by testing their approaches directly on CAR-NK cells. Their CAR-NK cells were modified to secrete a TGFβ receptor inhibitory peptide (UP01a) that disrupts the TGFβ signalling pathway in the TME. UP01a secretion reduced TGFβ-induced immunosuppressive factors and increased NK-cell infiltration in the TME, thereby enhancing the efficacy of CAR-NK cells against solid tumours in xenograft mouse models [194]. Another study introduces self-activating CAR-NK cells that block TGFβ1 signalling in the TME using a peptide called P6, which targets mesothelin in pancreatic tumours. P6 disrupts TGFβ1’s inhibitory effects by interfering with the SMAD2/3 pathway, restoring NK-cell activity, metabolism, and cytotoxicity. These CAR-NK cells demonstrate strong anti-tumour effects in both spheroid cultures and in vivo models, offering a promising strategy to overcome TGFβ1-mediated immune suppression and improve cancer immunotherapy [195]. Chaudry et al. followed a different approach. They demonstrated that co-transducing NK cells with a B7H3 CAR and a TGFβ dominant negative receptor (DNR) can preserve cytolytic function in the presence of exogenous TGFβ. However, this was only tested in vitro and thus more preclinical data are necessary to determine if this is a promising approach [102]. Moreover, in this hypoxic environment, cancer cells release high levels of adenosine triphosphate (ATP), which is degraded through ectoenzymes CD39 and CD73 to become adenosine (ADO) [196,197]. Adenosine then accumulates in the TME and negatively impacts immune cell function, including NK cells. More precisely, ADO is a strong purinergic mediator, which is able to signal NK cells to negatively regulate their anti-tumour function [198]. To avoid this, Chambers et al. genetically modified NK cells by introducing an anti-CD37 scFv CAR, enabling them to effectively eliminate lung cancer cells with high CD37 expression even under hypoxic conditions [108]. As mentioned above, the nutrition deficiency that CAR-NK cells face in the TME is another obstacle that must be overcome. Therefore, Nachef et al. proposed enhancing CAR-NK cells by engineering them with amino acid transporters to improve amino acid uptake and protect them from suppression in the nutrient-deficient TME [199]. Despite these promising approaches, a lot of research is still necessary to apply those strategies successfully and safely in the clinical situation.

### 3.2. Next-Generation Engineering

Improved CAR structures require creative modification strategies. A developing trend is the creation of next-generation CAR-NK cells, which are intended to produce more accurate and potent CARs with enhanced tumour recognition abilities. One approach involves so-called dual CAR-NK cells. They express two distinct CARs targeting different antigens, thereby improving their ability to recognize tumour cells, thus preventing immune escape [179,200]. Some dual CAR-NK systems have already shown initial success in haematological cancers, like the dual CAR-NK targeting BCMA and GPRC5D, which demonstrated notable results in killing multiple myeloma cells [201] and CD19/CD20 dual CAR-NK cells in acute lymphoblastic leukaemia [202]. Dual CARs, with their two different targets, can also overcome immune escape and address antigen heterogeneity, both common occurrences in solid tumours. Similarly, dual CAR-NKs targeting prostate cancer cells that express both the prostate-specific membrane antigen (PSMA) and the prostate stem cell antigen (PSCA), show that this is a promising approach for solid tumours [203]. Also, Eitler et al. developed dual CAR-NK cells targeting PD-L1 and HER2, which showed enhanced cytotoxicity in 3D spheroids and in in vivo solid tumour models, compared to single-target CAR-NK cells. They are able to effectively overcome immune escape caused by the loss or inaccessibility of a single target antigen [204]. A comparable system was developed by Zhi et al., but instead of HER2, they targeted NKG2D ligands and PD-L1 [205]. Dual CAR-NKs also can help to avoid trogocytosis. Trogocytosis is a biological process involving the uptake of cell membrane parts and its surface molecules by NK cells from antigen-presenting cells through direct cell–cell contact. Through this process, NK cells can acquire surface molecules from target cells. For CAR-NK cells, this means that CAR activation can enable the transfer of CAR-specific antigens from tumours to NK cells. This may reduce tumour antigen density, which weakens CAR-NK-cell binding. It can also cause self-recognition, sustained CAR-binding, and low reactivity. This is a major challenge for CAR-NK-cell therapy as it can endorse immune escape and ultimately leads to cancer relapse. To counteract this problem, Li et al. developed a dual CAR system consisting of a CAR targeting the cancer antigen and an inhibitory CAR, sending inhibitory signals to inhibit CAR-NK cells and other NK cells that present trogocytosed antigens. For this, they used an antigen-specific inhibitory KIR-based receptor (iCAR), which was able to inhibit a CAR-mediated trogocytic antigen-induced NK-cell fratricide [206] and, in the long run, prevent the eventually immune escape of cancer cells [19,179]. Also, Franzen et al. constructed a next-generation CAR construct targeting the tumour associated glycoprotein CEA, incorporating PD-1 checkpoint inhibitor and a CCR4 chemokine receptor, which does not induce fratricidal killing of CAR-NK92 cells [79].

Goodridge et al. developed a multi-targeting CAR-NK-cell system consisting of an anti-CD19 CAR, a non-cleavable CD15 receptor, and IL-15/IL-R15α to enhance the cancer-killing and persistence of CAR-NK cells [207]. Moreover, universal CARs are also an option. Mitwasi et al. developed a universal CAR that activates only when paired with α-GD2 (disialoganglioside 2) IgG4 to target GD2-positive cancer cells [208]. Additionally, Kang et al. established a universal CAR system using NK92 cells, more precisely a-Cot-NK92 cells that can target multiple antigens on tumour cells by using different conjugators, without the need for genetic re-engineering, with the use of cotinine-conjugated antibodies [209]. Nonetheless, there are only a few studies on the topic of dual CARs, multi-targeting CAR-NKs, or universal CAR-NKs targeting cancer cells. Therefore, additional preclinical studies are needed to investigate the full potential of CAR-NK cells in eliminating solid tumours.

Inducible CAR-NK systems are considered as next-generation CAR-NKs. These are CAR constructs that can be activated through specific stimuli. For instance, Wang et al. developed the inducible MyD88/Cd40 (iMC) system, which improves NK-cell cytotoxicity, cytokine secretion, and proliferation, through its activation. To further enhance proliferation and anti-tumour activity of CAR-NK cells, the authors coupled iMC activation with ectopic IL-15, and for safety reasons, they incorporated an orthogonal rapamycin-regulated Caspase-9 (iRC9) safety switch to eliminate the CAR-NKs if needed [210]. Inducible CARs also require further preclinical and clinical investigation to translate this into the clinics.

The CRISPR (clustered regularly interspaced short palindromic repeats) technology is an additional method for enhancing CAR-NK constructs that has recently become more widely recognized. CRISPR a gene-editing technology that allows the precise modification of DNA by using a programmable enzyme (Cas9) to cut and edit genetic sequences. It can be utilized to make modifications on CAR-NK cells that possibly can improve their lifespan and their tumour antigen-targeting ability and reduce the expression of cancer-related oncogenes [211]. In a recent study, Choi et al. used the CRISPR/Cas9 system to knockout CD70 in NK cells, while simultaneously using CD70-directed CAR-NK cells. The CD70-deletion CAR-NK cells with additional IL-15 expression showed potent cytotoxicity against several CD70-positive solid tumour cell lines and produced key cytokines [212]. The group around Guo et al. advanced this research further. They used CRISPR/Cas9 to not only knockout CD70 but also to knockout CBLB (casitas B-lineage lymphoma proto-oncogene B) and CISH (cytokine-inducible SH2-containing protein) to enhance the cytotoxicity of CD70-CAR-NK cells. They demonstrated that the CD70/CISH/CBLB triple-knockout CD70-CAR-NK cells enhanced anti-tumour activity against renal cell carcinoma (RCC) solid tumour cell lines and provided improved resistance to TGFβ-mediated inhibition, therefore not only avoiding immune escape but also immune inhibition with their novel approach [213]. Another interesting approach was conducted by Shankar et al., who utilized the CRISPR/Cas9 system to disrupt the KLRC1 gene, responsible for encoding the HLA-E-binding NKG2A receptor. Interestingly, they used the CRISPR technology also for inserting a GD2 (disialoganglioside)-targeting CAR into NK cells isolated from the human blood, with a virus-free genome editing workflow, which increases the safety of CAR-NK cells. This offers an alternative to other methods like viral-based or electroporation-based methods. This resulted in an 80% knockout efficiency and 23% targeted insertion with minimal off-target effects. The engineered KLRC1-GD2-CAR-NK cells showed strong viability, proliferation, and precise targeting of GD2^+^ tumours. They were also able to overcome HLA-E-based immune inhibition [214]. Overall, those studies show that the combination of CRISPR technology and CAR-NK cells is a promising method to not only modify CAR-NK cells to improve them but also to insert CAR constructs into NK cells in a safer way. However, again, further preclinical studies are needed to enable clinical use.

### 3.3. Emerging Targets

The identification of novel tumour-specific antigens is crucial for improving the precision and effectiveness of CAR-NK-cell therapies. Various approaches are being used to discover potential targets, including genomics, proteomics, transcriptomics, and single-cell sequencing analysis of normal and tumour tissues [215,216]. For solid tumours, various targets have been evaluated. HER2 is a well-established target that is overexpressed in several cancers, including breast, colon, and ovarian cancers [217]. HER2-CAR-NK cells, derived from PB NK cells or NK92 cells, have been investigated in various preclinical studies [95,218,219,220] and are currently being investigated in clinical trials (NCT03383978, NCT04319757). NKG2D ligands are often overexpressed in malignant cells, making them a good target for CAR-NK-cell therapies, which are currently investigated in several preclinical and clinical settings (NCT06478459, NCT05528341, NCT05213195) [221]. Another interesting new target is CD70, which has shown promising results in a variety of solid tumours [66,212,222]. Other targets under investigation include PSMA, which is expressed over 100 times higher in prostate tumour tissue than in normal prostate tissue (NCT03692663) [223], and the glycoprotein MUC1, which is responsible for creating a significant barrier that prevents drugs from binding to their targets on cancer cells (NCT02839954) [224]. Further targets that are currently investigated include the tumour differentiation antigen mesothelin for ovarian cancer [225], TROP2 (NCT06066424, NCT05922930) [206], ROBO1 (NCT03940820, NCT03941457) [130], EGFR [97], GCP3 (gamma-Tubulin complex protein 3) for hepatocellular carcinomas [111,112,131], B7H3 for glioblastoma [100,101], and members of the claudin family for gastric and pancreatic cancers (NCT06464965, NCT05410717) [120] (see also Table 4). Some clinical trials are also investigating 5T4 (trophoblast glycoprotein), a tumour-associated antigen expressed on the cell surface of most solid tumours (NCT05194709, NCT05137275). Söhngen et al. showed that CD24 is a promising novel target for urologic malignancies and demonstrated that CD24-CAR-NK cells especially highly target CD24^+^ cells, which ultimately resulted in enhanced anti-tumour activity [226]. Other more unusual targets that showed quite promising results when targeted by CAR-NK cells are ROR1 (receptor tyrosine kinase-like orphan receptor 1) in recurrent neuroblastoma [134]; EphA2 (ephrin type-A receptor 2) in paediatric sarcomas, including rhabdomyosarcoma, Ewing sarcoma, and osteosarcoma [227]; and CD44v6 [90] and HER3 in breast cancer cells [228].

## 4. Conclusions and Outlook

CAR-NK cells have emerged as a promising approach in the treatment of solid tumours. They offer advantages over CAR-T cells, including reduced neurotoxicity, allogenic potential and a reduced risk of GvHD [39]. The molecular basis of CAR-NK-cell function, preclinical and clinical developments, and difficulties in maximizing their therapeutic efficacy have been discussed here. Although clinical trials have shown promising results, several obstacles remain, including limited persistence, the suppressive TME, and the need for improved targeting techniques, which require innovative solutions. Particularly for solid tumours, where on-target/off-tumour toxicity is a major concern, logic-gated CARs may offer a potential solution. These are advanced CARs, designed for T cells, that use Boolean logic to require the recognition of multiple antigens for T-cell activation, reducing the risk of attacking normal tissues [229], for example, synthetic Notch receptors (synNotch) [230]. As they show quite promising results for T cells, this technique could be adapted for NK cells as well [231]. Another interesting approach for improving the efficacy of CAR therapy is the use of artificial intelligence (AI). AI techniques such as deep learning, natural language processing, and computer vision are being used to optimize CAR design, analyse cell morphology, and predict therapy efficacy and cytotoxicity. Integration of AI in CAR-NK-cell development could potentially improve treatment personalization and precision [232,233]. In addition, studies on vaccines with CAR-T cells have boosted the efficacy of CAR-T cells in eradicating cancer cells. It may therefore be possible to apply this strategy to CAR-NK cells [179]. Further, to improve the efficacy of CAR-NK cells, one group demonstrated that peptide-based CAR-NK cells, which have a peptide domain instead of an scFv targeting domain, could induce comparable anti-tumour activity in solid tumours and mitigate on-target/off-tumour toxicity compared to scFv-based CAR-NK cells in vivo [234]. Overall, CAR-NK-cell therapy represents a breakthrough in cancer immunotherapy, especially for solid tumours resistant to traditional treatments. While progress has been made, further research is needed to refine these therapies and enhance clinical outcomes. Advances in engineering, combination strategies, and personalized approaches could maximize their impact. With ongoing clinical trials and translational research, CAR-NK therapy holds great promise as a standard treatment for difficult-to-treat malignancies.

## Figures and Tables

**Figure 3 ijms-26-06290-f003:**
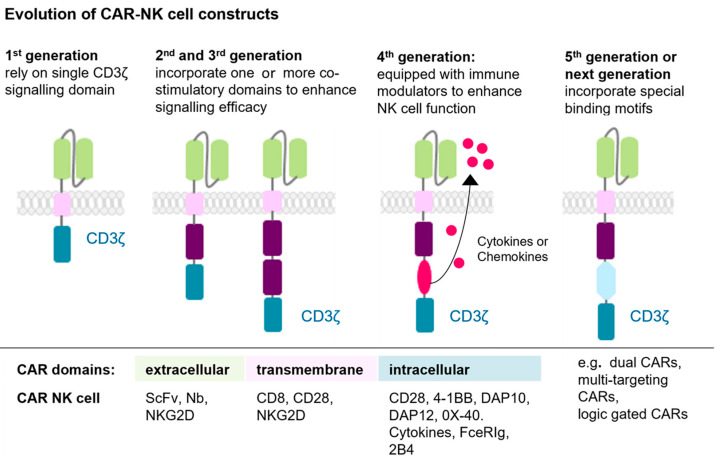
The evolution of CAR-NK cell constructs is shown. CAR generations evolved from a basic CD3ζ domain (first generation) to include one (second generation) or two (third generation) co-stimulatory domains, enhancing activation. Fourth-generation CARs added cytokine secreting elements, while fifth-generation constructs integrated specialized binding motifs for improved function. Next-generation constructs include: dual-CARs, multi-targeting CARs, inducible CARs, and logic-gated CARs (not separately depicted) [49,51,52]. This figure was generated with Motifolio.

**Table 3 ijms-26-06290-t003:** Clinical trials of CAR-NK cells in solid tumours (ClinicalTrials.gov, accessed on 11 June 2025).

Cancer Type	CAR Construct	Target Antigen	Cell Source	Clinical Trial Phase	NCT	Status	Study
Solid tumours	ROBO1 ^1^ CAR-NK cells	ROBO1	Human primary NK cells	Phase 1/2	NCT03940820	Unknown	Evaluation of the safety and effectiveness of ROBO1 CAR-NK cells in targeting ROBO1-expressing solid tumours
Pancreatic cancer	BICAR-NK cells ROBO1 CAR-NK cells	ROBO1	NK92 cell line	Phase 1/2	NCT03941457	Unknown	Evaluation of ROBO1 specific BiCAR-NK cells in patients with pancreatic cancer
MUC1 ^2^-positive solid tumours	Allogenic anti-MUC1 CAR-pNK cells	MUC1	NK92 cell line	Phase 1/2	NCT02839954	Unknown	Evaluation of the efficacy and safety of chimeric antigen receptor-modified pNK cells in MUC1 positive advanced refractory or relapsed solid tumour
Epithelial ovarian cancer	Mesothelin CAR-NK cells	Mesothelin	Human primary NK cells	Early phase 1	NCT03692637	Unknown	Investigation of the safety and efficacy of anti-mesothelin CAR-NK cells with epithelial ovarian cancer
Advanced gastric cancer, advanced pancreatic cancer	CAR- NK182 cells	Claudin18.2	Cord blood derived NK cells	Phase 1	NCT06464965	Recruiting	Evaluation the efficacy of CB CAR-NK182 in patients with advanced gastric cancer and advanced pancreatic cancer
Stage IV ovarian cancer, Testis cancer, Refractory endometrial cancer	Claudin6, GPC3 ^3^, Mesothelin, or AXL ^4^ targeting CAR-NK cells	Claudin6, GPC3, Mesothelin, AXL	Human primary NK cells	Phase1	NCT05410717	Recruiting	Evaluation the safety and preliminary efficacy of claudin6, GPC3, mesothelin, or AXL targeting CAR-NK cells in patients with claudin6, GPC3, mesothelin, or AXL-positive advanced solid tumours
Advanced or metastatic HER2 ^5^-expressing solid tumours	anti-HER2 NK cells	HER2	--	Phase 1	NCT04319757	Completed	A phase I, open label, dose escalation study of ACE1702 cell immunotherapy in subjects with advanced or metastatic HER2-expressing solid tumours
Non-resectable pancreatic cancer	NKG2D ^6^ CAR-NK cells		--	Early phase 1	NCT06478459	Recruiting	Evaluate the safety and anti-tumour efficacy of intratumoural NKG2D CAR-NK cell injection guided by EUS in the treatment of locally advanced pancreatic cancer.
Refractory metastatic colorectal cancer	NKG2D CAR-NK cells	NKG2D-ligand	--	Phase 1	NCT05213195	Recruiting	NKG2D CAR-NK cell therapy in patients with refractory metastatic colorectal cancer
Relapsed/refractory solid tumours	NKG2D-CAR NK92 cells		NK92 cell line	Phase 1	NCT05528341	Recruiting	Evaluation of the safety and effects of NKG2D-CAR-NK92 infusions in the treatment of relapsed/refractory solid tumours
Metastatic solid tumours	NKG2DL CAR-NK cells	NKG2D-ligand	Autologous or allogeneic NK cells	Phase 1	NCT03415100	Unknown	Evaluation the safety and feasibility of CAR-NK-cell treatment in subjects with metastatic solid tumours
Metastatic castration-resistant prostate cancer	PSMA ^7^ CAR-NK cells	PSMA	Human primary NK cells	Early phase 1	NCT03692663	Unknown	Evaluation the safety, tolerability and preliminary efficacy of TABP EIC in patients with metastatic castration-resistant prostate cancer
Advanced solid tumours	Anti-5T4 ^8^ CAR-NK cells	5T4	--	Early phase 1	NCT05194709	Unknown	Evaluation the safety, tolerability, initial efficacy and pharmacokinetics (PK) of anti-5T4 CAR-NK cells in patients with advanced solid tumours
Locally advanced or metastatic solid tumours	Anti-5T4 CAR-raNK cells	5T4	--	Early phase 1	NCT05137275	Unknown	Dose escalation and extension study to evaluate the safety, tolerability, and initial efficacy of anti-5T4 CAR-raNK-cell therapy in locally advanced or metastatic solid tumours
Recurrent or unresectable solid tumours	CAR-NK cells	--	--	Early phase 1	NCT06572956	Active	Clinical study on the safety and efficacy of CAR-T/CAR-NK cells in the treatment of recurrent refractory or unresectable solid tumours
Solid tumours with high TROP2 expression	TROP2 ^9^-CAR NK cells	TROP2	Cord blood derived NK cells	Phase 1	NCT06066424	Recruiting	Dose escalation and expansion study of TROP2 CAR engineered IL-15-transduced cord blood-derived NK cells in patients with advanced solid tumours
Platinum resistant ovarian cancer, mesonephric-like adenocarcinoma, pancreatic cancer	TROP2-CAR/IL-15-NK cells	TROP2	Cord blood derived NK cells	Phase 1/2	NCT05922930	Recruiting	Study of TROP2 CAR engineered IL-15-transduced cord blood-derived NK cells delivered intraperitoneally for the management of platinum resistant ovarian cancer, mesonephric-like adenocarcinoma, and pancreatic cancer
Hepatocellular carcinoma	SN301A	Glypican-3	--	Early phase 1	NCT06652243	Recruiting	Evaluation of the safety and efficacy of SN301A cell injection in the treatment of subjects with GPC3-positive advanced hepatocellular carcinoma
Ovarian epithelial carcinoma	SZ011 CAR-NK cells	--	--	Early phase 1	NCT05856643	Recruiting	Observation and investigation of the clinical safety and efficacy of SZ011 in the treatment of ovarian epithelial carcinoma
Pancreatic cancer	CAR-NK cells (CL-NK-001)	--	--	Phase 1	NCT06816823	Not yet recruiting	Evaluation of the safety, tolerability and efficacy of CAR-NK cells (CL-NK-001) in patients with locally advanced, metastatic, or recurrent pancreatic cancer
Non-small cell lung carcinoma	CCCR ^10^-NK cells	CCCR	NK92 cell line	Phase 1	NCT03656705	Completed	Evaluation of the safety and effects of CCCR-modified NK92(CCCR-NK92) infusions in previously treated advanced non-small cell lung carcinoma (NSCLC)

^1^ ROBO1: roundabout guidance receptor 1; ^2^ MUC1: mucin 1; ^3^ GCP3: glypican-3; ^4^ AXL: tyrosine–protein kinase receptor AXL; ^5^ HER2: human epidermal growth factor receptor 2; ^6^ NKG2D: natural killer group 2 member D; ^7^ PSMA: prostate-specific membrane antigen; ^8^ 5T4: trophoblast glycoprotein; ^9^ TROP2: trophoblast antigen 2; ^10^ CCCR: chimeric costimulatory converting receptor.

**Table 4 ijms-26-06290-t004:** Overview of recent CAR-NK cell combination therapies.

Cancer Type	CAR Construct	Target Antigen	Cell Source	Drug Category	Drugs	Reference (PMDI/NTC + Status)	Intention
Castration-resistant prostate cancer	CAR NK92 cells	PSMA ^1^	NK92 cell line	Immune checkpoint inhibitors	Anti-PD-L1 ^2^ mAb; Atezolizumab	[126]	Anti-PD-L1 mAb markedly boosted the anti-tumour efficacy of CAR-NK92 cells
Recurrent HER2 ^3^-positive glioblastoma	NK92/5.28.z cells	HER2	NK92 cell line	Immune checkpoint inhibitors	Anti-PD-1 Ab; Ezabenlimab	NCT03383978 Active, not recruiting	Intracranial injection of NK92/5.28.z cells in combination with intravenous Ezabenlimab in patients with recurrent HER2-positive glioblastoma
Nasopharyngeal cancer mouse model	CAR pNK cells	PD-L1	Haematopoietic stem cells	Immune checkpoint inhibitors	Anti-PD-1 Ab: Nivolumab	[127]	CAR pNK cells and Nivolumab resulted in a synergistic anti-solid tumour response
Recurrent/metastatic gastric cancer, head and neck cancer	PD-L1 CAR-NK cells (PD-L1 t-haNK)	PD-L1	--	Immune checkpoint inhibitors	Pembrolizumab, N-803	NCT04847466, recruiting	Testing effectiveness of irradiated PD-L1 CAR-NK cells, combined with Pembrolizumab and N-803, in people with advanced forms of gastric or head and neck cancer
Anaplastic thyroid carcinoma	NKG2D ^4^ CAR-NK cells	NKG2D ligands	--	Immune checkpoint inhibitors	Anti-PD-1 Ab	NCT06856278, not yet recruiting, new	Clinical study of NKG2D CAR-NK combined with PD-1 monoclonal antibody in the treatment of ATC
Colorectal cancer	TROP2 ^5^-CAR-NK cells	TROP2	Cord-blood derived NK cells	Monoclonal Ab	Cetuximab	NCT06358430, recruiting	Dose escalation and expansion study of TROP2 CAR engineered IL-15-transduced cord blood-derived NK cells in combination with Cetuximab in patient with colorectal cancer
Ovarian cancer	anti-CD44 CAR cells	CD44	NK92 cell line	Chemotherapy	Cisplatin	[128]	Treatment with CD44NK and cisplatin exhibited greater anti-tumour activity compared to sequential administration
Ovarian cancer	anti-CD133-CAR cells	CD133	NK92 cell line	Chemotherapy	Cisplatin	[129]	Cisplatin followed by CAR-NK-cell treatment resulted in the strongest killing effect
Relapsed/refractory non-small lung cancer	anti-Trop2 U-CAR-NK cells	TROP2	--	Chemotherapy	--	NCT06454890, not yet recruiting	Evaluating the efficacy and safety of Anti-Trop2 universal CAR-NK (U-CAR-NK) cells therapy combined with chemotherapy for relapsed/refractory non-small cell lung cancer
Pancreatic cancer	NKG2D CAR-NK cells	NKG2D ligands	--	Chemotherapy	--	NCT06503497, recruiting	Second-line chemotherapy sequential NKG2D CAR-NK-cell therapy for Pancreatic Cancer
Advanced Renal cell carcinoma, Advanced mesothelioma, Advanced osteosarcoma	CAR.70/IL-15-transduced CB-derived NK cells	CD70	Cord-blood derived NK cells	Chemotherapy	Cyclophosphamide, Fludarabine phosphate	NCT05703854, recruiting	Study of CAR.70-engineered IL-15-transduced cord blood-derived NK cells in conjunction with lymphodepleting chemotherapy for the management of advanced renal cell carcinoma, mesothelioma and osteosarcoma
Pancreatic carcinoma	ROBO1 ^6^ specific CAR-NK cells	ROBO1	NK92 cell line	Radiotherapy	^125^I Seed Brachytherapy	[130]	CAR-NK cells achieved the strongest killing effect, enhancing the efficacy of ^125^I seed brachytherapy in an orthotropic model
Hepatocellular carcinoma	CXCR2 ^7^-armed GPC3-targeting CAR-NK92 cells	Glypican-3	NK92 cell line	Radiotherapy	8 Gy	[131]	Evidence that irradiation could efficiently enhance the anti-tumour effect of CAR-NK cells in solid tumour model
Pancreatic cancer	CAR-NK92 cells	Mesothelin, cGAMP	NK92 cell line	STING ^8^ agonists	cyclic GMP-AMP	[109]	cGAMP could directly activate NK cells and enhance the sensitivity of pancreatic cancer cells to NK-cell cytotoxicity
Malignant pleural mesothelioma (patient samples)	anti-MSLN ^9^ CAR-NK cells	Mesothelin	--	STING agonists	TAK-676	[132]	NK-cell therapies (CAR-NKs) benefit from STING agonist enhancement of NK-cell migration and killing
Colorectal cancer	EpCAM ^10^-CAR-NK92 cells	EpCAm	NK92 cell line	Tyrosine-kinase inhibitor	Regorafenib	[117]	Improve the therapeutic effectiveness of CAR-engineered immune effector cells against solid tumours
Renal cell carcinoma	CAR-NK92 cells	EGFR ^11^	NK92 cell line	Tyrosine-kinase inhibitor	Cabozantinib	[118]	Boost the cytotoxicity of CAR-NK92 cells against RCC cells
Advanced renal cell carcinoma	CAR-NK92 cells	CAIX ^12^	NK92 cell line	Protease inhibitor	Bortezomib	[133]	Bortezomib enhances the efficacy of CAR-NK92 cells against RCC both in vitro and in vivo.
Glioblastoma	EGFR-CAR NK cells	EGFR	Peri-pheral-blood derived NK cells	Oncolytic virus	OV-IL-15C	[99]	The combination of an oncolytic virus expressing the IL-15/IL-15Rα complex and frozen, ready-to-use EGFR-CAR NK cells elicit strong anti-tumour responses in glioblastoma
Breast cancer brain metastasis	EGFR-CAR NK92 cells	EGFR	NK92 cell line	Oncolytic virus	Oncolytic herpes simplex virus oHSV-1	[93]	Demonstration that regional administration of EGFR-CAR NK92 cells combined with oHSV-1 therapy is a potentially promising strategy to treat breast cancer brain metastasis
Recurrent/metastatic neuroblastoma	anti-ROR1 ^13^ CAR-NK cells	ROR1	-	Oncolytic virus	Oncolytic herpes simplex virus C021	[134]	The combination of an oncolytic virus expressing hL-21 and anti-ROR1-CAR-NK cells is an effective immunotherapeutic approach
Lung cancer	CAR-NK cells	B7-H3	NK92MI cell line	Photothermal therapy	Near infrared (NIR-II)	[135]	Synergistic CAR-NK immunotherapy is designed to specifically eliminate any residual tumour cells following PTT

^1^ PSMA: prostate-specific membrane antigen; ^2^ PD-L: programmed death ligand; ^3^ HER2: human epidermal growth factor receptor 2; ^4^ NKG2D: natural killer group 2 member D; ^5^ TROP2: trophoblast antigen 2; ^6^ ROBO1: roundabout guidance receptor 1; ^7^ CXCR2: C-X-C motif chemokine receptor 2; ^8^ STING: stimulator of interferon genes; ^9^ MSLN: mesothelin; ^10^ EpCAM: epithelial cell adhesion molecule; ^11^ EGFR: epidermal growth factor receptor; ^12^ CAIX: carbonic anhydrase IX; ^13^ ROR1: receptor tyrosine kinase-like orphan receptor 1.

## Data Availability

Not applicable.

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
