# Peer review of "Rendering NK Cells Antigen-Specific for the Therapy of Solid Tumours"

_ijms, 2025, doi:10.3390/ijms26136290_

Round 1
Reviewer 1 Report
Comments and Suggestions for Authors
L54-55 I understand that CAR-T is not the main topic of this paper. However, you mention them on several occasions. You should consider dedicating brief lines to describe how they work, so that the reader can understand the advantages of the CAR-NK
Watching table 1, it comes to my mind that I would have liked to see a table where you also compare the advantages of CAR-NK over CAR-T
In figure 2 you name "stimulatory receptors" and in the text you call them activating receptors. consider to homogenize the text
These are only suggestions, in general I consider your work of a high value.
Author Response
Reviewer 1
First of all, we thank the reviewer for reading and evaluating our manuscript.
L54-55 I understand that CAR-T is not the main topic of this paper. However, you mention them on several occasions. You should consider dedicating brief lines to describe how they work, so that the reader can understand the advantages of the CAR-NK
Answer: Thank you for mentioning this, we added some brief lines to describe how they work (line 281-298).
Watching table 1, it comes to my mind that I would have liked to see a table where you also compare the advantages of CAR-NK over CAR-T
Answer: We added a table to compare the advantages of CAR-NK cells over CAR-T cells (new table 1; line 307).
In figure 2 you name "stimulatory receptors" and in the text you call them activating receptors. consider to homogenize the text
Answer: We have adjusted the naming of the receptors into “activating receptors” to homogenize the text.
These are only suggestions, in general I consider your work of a high value.
Answer: Thank you for your kind evaluation.
Reviewer 2 Report
Comments and Suggestions for Authors
The manuscript by Doeppner et al. reviews the application and progress of NK cells as antigen-specific therapy for solid tumors. The background of NK cells in the cancer microenvironment and their roles in cancer immunotherapy is thoroughly introduced. The progress of using CAR-NK cells for solid tumor suppression is discussed, with their pros and cons, along with detailed and up-to-date preclinical and clinical evidence. The combination of CAR-NK therapies with various conventional treatments is also reviewed in detail. Lastly, the authors list the challenges and future directions of CAR-NK-based antitumor therapies.
Immunotherapy of cancers holds the potential for long-term remission, the ability to overcome resistance to other treatments, and the possibility of preventing cancer recurrence, in which CAR-NK cells offer promising avenues for cancer immunotherapy. This is a very comprehensive review manuscript covering aspects from molecular mechanisms to clinical research. This review provides valuable insights for both researchers and clinicians working to advance NK cell-based cancer therapies.
The background is presented with figures for easy reading, and research progress is well summarized with tables. The language is clear, and figures/tables are of good quality. I suggest publishing the manuscript in the current version.
Author Response
Reviewer 2
The manuscript by Doeppner et al. reviews the application and progress of NK cells as antigen-specific therapy for solid tumors. The background of NK cells in the cancer microenvironment and their roles in cancer immunotherapy is thoroughly introduced. The progress of using CAR-NK cells for solid tumor suppression is discussed, with their pros and cons, along with detailed and up-to-date preclinical and clinical evidence. The combination of CAR-NK therapies with various conventional treatments is also reviewed in detail. Lastly, the authors list the challenges and future directions of CAR-NK-based antitumor therapies.
Immunotherapy of cancers holds the potential for long-term remission, the ability to overcome resistance to other treatments, and the possibility of preventing cancer recurrence, in which CAR-NK cells offer promising avenues for cancer immunotherapy. This is a very comprehensive review manuscript covering aspects from molecular mechanisms to clinical research. This review provides valuable insights for both researchers and clinicians working to advance NK cell-based cancer therapies.
The background is presented with figures for easy reading, and research progress is well summarized with tables. The language is clear, and figures/tables are of good quality. I suggest publishing the manuscript in the current version.
Answer: Thank you for this very kind evaluation.